# Influences of Successive Exposure to Bleaching and Fluoride Preparations on the Surface Hardness and Roughness of the Aged Resin Composite Restoratives

**DOI:** 10.3390/medicina56090476

**Published:** 2020-09-16

**Authors:** Khalid M. Abdelaziz, Shugufta Mir, Shafait Ullah Khateeb, Suheel M. Baba, Saud S. Alshahrani, Eman A. Alshahrani, Zahra A. Alsafi

**Affiliations:** 1Department of Restorative Dental Sciences, College of Dentistry, King Khalid University, Abha 61471, Saudi Arabia; shmir@kku.edu.sa (S.M.); skhateeb@kku.edu.sa (S.U.K.); baba@kku.edu.sa (S.M.B.); 2Intern, College of Dentistry, King Khalid University, Abha 61471, Saudi Arabia; Saoood26@hotmail.com (S.S.A.); Ashahranieman9@gmail.com (E.A.A.); zahralsafi1@gmail.com (Z.A.A.)

**Keywords:** aging, bleaching agents, surface hardness, surface roughness, resin composite, topical fluoride

## Abstract

*Background and Objectives:* Surfaces of composite restorations are adversely affected upon bleaching and topical fluoride application. Such a procedure is normally carried out in the presence of restorations already serving in a different oral environment, although previous in vitro studies only considered the freshly-prepared composite specimens for assessment. The current study accordingly aimed to evaluate both the surface hardness and roughness of aged composite restoratives following their successive exposure to bleaching and topical fluoride preparations. *Materials and Methods:* Disc specimens were prepared from micro-hybrid, nano-filled, flowable and bulk-fill resin composites (groups 1–4, *n* = 60 each). All specimens were subjected to artificial aging before their intermittent exposure to surface treatment with: none (control), bleach or topical fluoride (subgroups 1–3, *n* = 20). All surface treatments were interrupted with two periods of 5000 thermal cycles. Specimens’ surfaces were then tested for both surface hardness (Vickers hardness number (VHN), *n* = 10) and roughness (Ra, *n* = 10). The collected VHNs and Ras were statistically analyzed using two-way ANOVA and Tukey’s comparisons at α = 0.05 to confirm the significance of differences between subgroups. *Results:* None of the tested composites showed differences in surface hardness and roughness between the bleached and the non-treated specimens (*p* > 0.05), but the bleached flowable composite specimens only were rougher than their control (*p* < 0.000126). In comparison to the control, fluoride treatment not only reduced the surface hardness of both micro-hybrid (*p* = 0.000129) and flowable (*p* = 0.0029) composites, but also increased the surface roughness of all tested composites (*p* < 0.05). *Conclusion:* Aged composite restoratives provide minimal surface alterations on successive bleaching and fluoride applications. Flowable resin composite is the most affected by such procedures. Although bleaching seems safe for other types of composites, the successive fluoride application could deteriorate the aged surfaces of the tested resin composites.

## 1. Introduction

The functionality and stability of dental restorations usually rely on their structural integrity and sound interfacial bonding. Different materials have been used for direct tooth restorations and many of them showed the ability to achieve that task. However certain characteristics such as the metallic, non-esthetic nature of dental amalgam and the lower durability of glass-ionomer restorative materials could limit the range of their clinical applications [1,2]. Nowadays esthetic restorative materials are an important part of modern dentistry, and based on this point resin composites get their popularity. These materials are mainly composed of polymerizing-resin and dispersed filler phases coupled together with a bi-functional bonding phase. The compositional ratios and the structural stabilities of these components, and their interfacial bonding, help achieve the characteristics the manufacturer claims and govern the clinical performance of the composite material [3].

Contemporary resin composites usually show acceptable clinical durability and express surface and optical characteristics that nearly mimic those of the natural tooth structure. The utilization of nano-sized fillers would offer resin composites with higher translucency and maintain physico-mechanical properties equivalent to those of hybrid composites [4]. Flowable composites, in spite of their lower filler loading, get increased popularity in many clinical applications in response to their ease of application and occasionally for their self-adhesive ability [5,6]. Recently, bulk-fill resin composites were launched with the advantages of shortened application times, acceptable degrees of polymerization and lower polymerization shrinkage stresses in comparison to conventional types [7].

Surface hardness, on one hand, reflects the ability of restorative composite resin materials to resist the in-service mechanical degradation (Wear); however, this character could be affected by the size and the amount of the material’s filler contents [8,9,10]. Surface roughness of dental composites, on the other hand, shows a great influence on the adhesion and retention of dental plaque that in turn increases the potential risks of dental caries and periodontal diseases [11,12]. Surface roughness was also found to affect the color, the gloss and the staining susceptibility of resin composite restoratives [13]. Although surface hardness and roughness of resin composite restorations are mainly affected by the operator’s finishing and polishing skills, other factors related to patients’ dietary behaviors and caries-preventive measures could also influence restorations’ surface characteristics [14,15].

Tooth bleaching is currently known as an effective, safe and conservative esthetic rehabilitation technique [16,17]; however, the commonly utilized bleaching products have been reported to adversely affect the surfaces of resin-based restoratives. The noticed negative effects could vary according not only to the compositions of both restorative and bleaching materials, but also to the frequency and the duration of their contact [18]. Some researchers reported controversial findings when resin composites were exposed to home-bleaching products, while others [19,20] reported significant increase in the surface roughness of bleached resin composite surfaces. On the other hand, topical fluorides were reported to cause chemical degradation in resin composites’ surfaces and decrease their wear resistance, although the severity of the noticed degradation was dependent on the types of tested resin composite and fluoride materials [21]. Some in vitro studies also confirmed the susceptibility of resin composite surfaces to changes in their morphologies in response to fluoride surface treatment [22].

The aforementioned information revealed obvious negative effects of both bleaching and topical fluoride products on the surfaces of resin composite restoratives; however, few studies addressed the expected alterations in the surface properties of contemporary resin composites following their successive exposure to bleaching and topical fluoride products [23,24]. It was also noticed that all the reviewed in vitro studies did utilize freshly-prepared resin composite specimens [23,24], although in real practice both bleaching and fluoride application procedures are usually conducted in the presence of already serving composite restorations. Those restorations, all the time, are subjected to different oral environments, including cyclic fluid sorption, temperature and load changes [25,26]. Therefore, the current in vitro study aimed to evaluate the surface hardness and roughness of different types of aged contemporary resin composites when subjected to successive applications of bleaching and topical fluoride products. The null hypothesis accordingly suggested no effect of the utilized bleaching and fluoride preparations on composites’ surface hardness and roughness.

## 2. Materials and Methods

A total of 240 disc specimens, 7 mm in diameter and 3 mm thick, were constructed in 4 groups (*n* = 60 each) from micro-hybrid, nano-filled and flowable conventional resin composites (Filtek Z250, Filtek 350XT and Filtek 350XT Flowable, 3M ESPE, St. Paul, MN, USA) in addition to a paste-like bulk-fill resin composite (Filtek Bulk-fill, 3M ESPE, St. Paul, MN, USA). The brand names, descriptions and manufactures of the utilized materials are listed in Table 1. For all groups, shade A2 of the selected materials was used to construct the needed specimens in a custom-made split Teflon mold against 2 celluloid strips (Hawe Transparent Strips, Kerr, Orange, CA, USA) supported with glass plates. In groups 1–3 (G1, G2 and G3), resin composites were adapted into the mold in 2 increments; each was cured for 20 s using the Elipar S10 light curing unit (3M ESPE, Seefeld, Germany) with a light intensity of 1200 mW/cm^2^ at a wave length of 430–480 nm. To ensure standardized curing of the top composite layer, the light tip was kept in direct contact with the glass slab surface. In group 4 (G4), specimens of bulk-fill resin composites were adapted into the mold cavity in one bulk and cured through the glass plate on the top of the mold. After curing, specimens were released out of the mold and the top surface of each was notch marked for easy identification. To simulate the clinical situation, all specimens were then finished using fine composite finishing diamonds (Brasseler USA Dental, Savannah, GA) and serially polished with 2 Sof-Lex discs (#1982SF and 2382SF, 3M ESPE, St. Paul, MN, USA), each used for for 10 s, under copious air–water cooling. The process of simulated aging was then started by storing all specimens in water at 37 ± 1 °C for 24 h (FUNCTION Line incubator, Thermo Electronic Inc., Lagenselbold, Germany), followed by thermocycling at 5–55 °C for 10,000 cycles with 1 min dwell time (MSCT-1 Thermocyler, São Carlos, SP, Brazil) that almost equivalent to 1 year of aging [27].

The top surfaces of the aged specimens in each group were then treated in 3 different subgroups (*n* = 20), as shown in Figure 1. In subgroup 1 (SG1), all specimens were subjected to 2 extra episodes of thermocycling each for 5000 cycles alternating with storage for 24 h in water at 37 ± 1 °C, and their surfaces received no surface treatment for control purpose. Subgroup 2 (SG2) specimens were first subjected to thorough air-drying for 60 s before we coated each of their top surfaces with a 0.5–1 mm thick layer of a chemically-activated in-office bleaching gel (Opalescence Boost PF, Ultradent Products Inc, South Jordan, UT, USA) using the recommended syringe tips. The applied bleaching preparation was kept on specimens’ surfaces for 60 min (3 times X 20 min interval) before suctioning the applied gel and washing the composite surfaces with air–water spray for 15 s. The same bleaching procedure was repeated for the same specimens for 2 more times alternated with periods of 5000 thermal cycles each representing 6 m of intra-oral service [27,28,29]. Subgroup 3 (SG3) specimens were first subjected to thorough air-drying for 60 s before coating their top surfaces with acidulated phosphate fluoride gel that releases 1.23% fluoride ions (APF gel, Deepak Inc., Miami, FL, USA). The applied fluoride preparation was left undisturbed on specimens’ surfaces for 4 min [30] before washing out using air–water spray for 15 s. The same procedure was repeated 2 more times for the same specimens; those were subjected to 5000 thermal cycling in between the application times representing 6 m of intra-oral service.

Half the number of specimens in each subgroup (*n* = 10) were then utilized to evaluate the Vickers hardness number (VHN)—composites’ surface hardness. Five indentations were conducted for each specimen on a METKON MH microhardness tester (MH series, MITKON, Bursa, Turkey) using a load of 50 g for 30 s dwell time. The VHN measurement was immediately calculated with the aid of built-in computer and the mean for each specimen was then calculated and utilized for statistical analysis. The other 10 specimens of each subgroup were used to assess the changes in surface roughness. The roughness average (Ra) was assessed by a stylus-based surface profiler (Alpha-Step IQ, KLA Tencor, San Jose, CA, USA). Each specimen’s surface was subjected to 5 3 mm long stylus tracks in different directions. The Ra measurement was automatically calculated in micrometers with the aid of the machine accompanied software. The mean Ra was then calculated for each specimen before further statistical analysis took place. The collected data were statistically analyzed using both 2-way ANOVA and Tukey’s pairwise comparisons at α = 0.05 to stand on the significance of differences detected between test subgroups.

## 3. Results

The calculated means of VHNs and standard deviations (SD) of resin composites in different subgroups are displayed in Table 2, while the means and SD of Ra values of resin composites in different subgroups are shown in Table 3.

### 3.1. Surface Hardness

The two-way ANOVA analysis of the collected VHN data indicated significant differences between test groups (resin composites, *p* < 0.0001) and between subgroups (surface treatment, *p* < 0.0001) in addition to a significant interaction between both variables (*p* = 0.0011).

For all test groups, Tukey’s comparisons indicated no differences between the VHNs of non-treated (SG1) and bleached (SG2) composite specimens (*p* > 0.05). In G1 (micro-hybrid composite) and G3 (flowable composite), the fluoride-treated surfaces (SG3) showed lower VHNs in comparison to the non-treated specimens (SG1), while in G2 (Nano-fill composite) and G4 (Bulk-fill composite), the fluoride-treated surfaces were as hard as the non-treated surfaces in SG1 (*p* > 0.05).

Within different test subgroups, the non-treated surfaces (SG1) of the micro-hybrid composite (G1) showed higher VHNs than the non-treated surfaces of other composite groups (*p* < 0.05). In SG2 (Bleached composite surfaces) the micro-hybrid composite (G1) also showed higher VHNs than other groups, although the flowable (G3) and bulk-fill (G4) composites exhibited comparable values (*p* = 1.000). In SG3 (fluoride-treated composite surfaces), comparable VHN values were detected for micro-hybrid (G1) and nano-filled (G2) composites and for flowable (G3) and bulk-fill (G4) composites (*p* > 0.05) respectively.

### 3.2. Surface Roughness

Analyzing the collected Ra values using two-way ANOVA indicated significant differences between test groups (resin composites, *p* < 0.0001) and between subgroups (surface treatment, *p* < 0.0001). A significant interaction between both variables (*p* < 0.0001) was also detected.

In all test groups, no differences in the Ra values were noticed between non-treated (SG1) and bleached (SG2) composite specimens (Tukey’s, *p* > 0.05). The only exception was noticed in G3 (flowable composite) where the bleached surfaces (SG2) exhibited higher Ra values than the non-treated (SG1) ones (*p* < 0.05). The fluoride treated specimens (SG3) in all test groups showed higher Ra values in comparison to both non-treated and bleached specimens (*p* < 0.05).

In SG1 (non-treated composite surfaces), no differences were noticed between different resin composite groups (*p* > 0.05). In SG2 (bleached composite surfaces), flowable composite (G3) showed higher Ra values than other groups of resin composites (*p* < 0.05). In SG3 (fluoride-treated composite surfaces), flowable composite (G3) owned the highest Ra value among all test groups. However, both nano-filled (G2) and bulk-fill (G4) composites exhibited comparable Ra values (*p* > 0.05) higher than the micro-hybrid type (G1).

## 4. Discussion

Restoring defective teeth using contemporary resin composites currently represents a routine approach in everyday dental practice. Contemporary resin composites showed great advancements not only in their compositional formulations but also in their application techniques. These materials can mimic, to a great extent, both surface and optical tooth characteristics [3,4]. However, these features could be adversely affected by certain in-office procedures, such as bleaching and topical fluoride applications [14,15].

Normally, dental bleaching is a short-term approach to improving teeth esthetics. Several in-office and at-home bleaching products are currently marketed; however, both carbamide and hydrogen peroxide-based gels seem common for in-office bleaching procedures. Although bleaching treatment appears non-invasive for the patients, some authors indicated adverse effects of this procedure not only on oral and tooth tissues, but also on the existing dental restorations. These effects have been mainly related to both potency and acidity of the utilized products [31]. Some manufacturers, accordingly, claimed the reduction of these drawbacks by either reducing the concentrations of the active ingredients or neutralizing the pHs of their products [32]. Therefore, the recently introduced Opalescence Boost PF (Ultradent) with a neutral pH (7) was selected for this study.

Infrequent professional topical fluoride application (for 1–4 min at 3–12 m intervals) is still a common preventive procedure, especially in pediatric and high caries-risk patients. Some orthodontists also utilize it to help minimize the chances of tooth decay during the long period of treatment [33]. Topical fluorides strengthen teeth already present in the mouth, making them more decay resistant. On the contrary, some erosive and softening effects of the topical fluoride preparations have been noticed on surfaces of both tooth enamel and some existing dental restorations [34,35]. Accordingly, the APF foam containing 1.23% fluoride (Gelato APF gel, Deepak) was selected for this study due to convenient application, fluoride uptake comparable to that of the gel preparations and reduced chance of systemic ingestion [36].

In comparison to dental amalgam and glass-ionomer, resin composites are the most commonly used direct esthetic tooth restorative material. These materials own a wide range of clinical applications in addition to the acceptable mechanical and esthetic durability [3,37]. In this study, resin composites were selected in response to their popularity in daily dental practices. In this study, all restorative materials (Filtek Z250, Filtek Z350 XT, Filtek 350 XT Flowable and Filtek Bulk-fill; 3M ESPE) were selected—in spite of the existing compositional differences (Table 1)—to have the same manufacturing technology for fair comparisons. On the other hand, aging of cured resin composites was known to improve a material’s degree of conversion and subsequently its physico-mechanical properties; those, in turn, would help with resistance to the different intraoral deteriorating environments [38]. Therefore, the current study aimed to assess the influences of successive applications of both bleaching and topical fluoride preparations on both surface hardness and roughness via aged resin composite specimens.

The results indicated insignificant reductions in the VHNs of bleached composite specimens, in comparison to the control, of different groups (Table 2). These results came in disagreement with some previous studies [39,40] which indicated significant reductions in the surface hardness of bleached resin composites. One study [39] revealed an adverse effect of the bleaching procedure on the surface hardness of bulk-fill resin composites. Another study [40] also indicated a reduction in the surface hardness of a nano-filled resin composite when bleached with an in-office carbamide peroxide-based preparation for 30 min/W for 3 weeks. Although these results were related to the possible degradation of the resin matrix and the resin–filler bond [41], the obvious contradiction with the results of the current study could be explained by the differences in the formulation of the utilized resin composites and bleaching preparations (Table 1). Moreover, aging normally increases the degree of resin matrix conversion and helps elute the residual monomer out of the material [38]. These outcomes could improve the resistance of the tested resin composites to the action of the bleaching ingredients.

Contrary to the aforementioned studies, others can support the findings of the current study. Results of Bicer et al. [42] showed no significant effect of carbamide or hydrogen peroxide-based bleaching systems on the surface hardness of either nano-hybrid or nano-flowable resin composites. They related their findings to the possible buffering and diluting effects of saliva and water on the concentrations and pHs of the utilized peroxide preparations. In spite of the different brands of resin composites, another study [43] indicated no change in surface hardness for both nano-filled and nano-hybrid bulk fill resin composites. Authors of that study related their finding to the large surface areas of the nano-sized filler particles that would logically result in higher chances for the indenter to hit the filler particles themselves at the time of hardness testing. Kwon et al. [44], in addition, reported no significant differences in surface hardness of the nano-filled resin composite (Filtek-Z350) specimens—the same as that used in the current study—following their contact with distilled water, carbamide peroxide and hydrogen peroxide gels. They explained their findings based on the similar softening effect of all surface treatments, either water or bleaching agents, on a composite’s resin matrix.

Results of the current study showed different effects of fluoride gel application on the surface hardness of the tested resin composites (Table 2). Both fluoride-treated nano-filled and bulk-fill resin composites showed VHNs comparable to those of their controls (aged with no surface treatment), while both flowable and micro-hybrid composites showed lower hardness values than their controls. Many studies [45,46,47] reported controversial effects of the acidulated phosphate fluoride (APF) preparations on composites’ surface hardness and some of their authors [47] related that controversy mainly to the differences in composites’ filler contents (loading, size and type) and to the composite-APF contact times. The APF gel was found to damage the resin matrix, the inorganic fillers and the resin-filler interfaces, leading to separation of the filler particles [48]. Therefore, results of the current study could be explained based on the compositions of the used resin composites, as shown in Table 1. Both nano-filled and bulk-fill resin composites of the current study utilize the same filler systems (type and size) and a significant part of this systems is composed of both zirconia and ytterbium trifluoride particles; those could resist the etching effect of the applied fluoride preparation. At the same time, the filler system of micro-hybrid resin composites contains larger fillers with significant contributions from silica particles which are easily affected by the applied fluoride treatment.

Although the flowable composite contains nearly the same type and size of fillers as those of the nano-filled composite (Table 1), the lower filler loading allows for a significant action of the fluoride gel preparation on the hardness of the resin matrix itself. This explanation is supported by the findings of Mazaheri et al. [49], which indicated maximal reduction in surface hardness values of both unfilled and low-filled resin materials following their surface treatments with the APF. They explained their results as the acidity of APF helping water to bind with the resin matrix, increasing its softness and probably offering a chance for its hydrolysis. On the other hand, results of Rashidian et al. [6] indicated no change in the surface hardness of non-aged flowable resin composites (Tetric N-Flow, Ivoclar-Vivadent; PermaFlo, Ultradent and Denfil, Vericom) following single 4 min exposures to APF. The difference between this finding and that of the current study could be related not only to the differences in the utilized materials (Filtek Z350 XT Flowable, 3M ESPE) and their aging, but also to the different fluoride application protocol (successive application).

The surface roughness for a resin composite restorative usually has an impact on the possible composite discoloration, retention of dental plaque and gingival irritation [49]. Some authors [50] reported that Ra of 0.2 µm is the critically acceptable value of a restoration’s surface roughness. Records of the current study showed that resin composite surfaces in all test subgroups exhibited Ras lower than the critical value (Table 3). These findings could be a result of the applied aging procedure. Aging of the resin composite normally leads to an increase in the degree of resin polymerization [8] that in turns reduces the possibility of its erosion in water [51] or in the presence of oxidizing and acidic chemicals [52]. The results of several studies which have been conducted on freshly-prepared composite specimens indicated Ras higher than the clinically acceptable roughness value. Hafez et al. [23] reported roughness values of 106.7 ± 2.1–126.1 ± 5.3 nm in resin composite specimens subjected to double applications of 35–38% H_2_O_2_ from in-office bleaching agents after their storage in water for only 24 h. The increase in surface roughness as a result of the bleaching process, in addition, was primarily related to the type and shade of the tested resin composite. Dionysopoulos and Koliniotou-Koumpia [24] also reported an average surface roughness of 0.21 ± 0.03–0.27 ± 0.03 µm in composite specimens after their storage in artificial saliva for only 24 h. Surfaces of the same composite materials showed even higher Ra values, 0.23 ± 0.04–0.59 ± 0.06 µm, following APF application.

Referring back to the results of the current study, the noticed roughness of the non-treated composite specimens of all groups could be a result of the finishing and aging processes. Contact with either water or saliva could, at the same time, cause chemical degradation of resin composite surfaces, leading to an increase in their roughness values. Giannini et al. [51] indicated possible breakdown of filler–resin bonds in contact with storing water, which in turn facilitates the debonding/loss of filler particles. However, the varied hydrolytic stability of the coupling agent in addition to fillers’ type, size and loading could result in different roughness values of different resin composite restoratives.

Most of the bleached types of resin composites recorded Ra values comparable to those of the non-treated surfaces. In spite of the possible effect of aging [8], only flowable composite surfaces exhibited higher roughness on bleaching than did the non-treated flowable composite and all other types of resin composites (Table 3). These findings indicated the significant effect of the utilized bleaching agent on the surface topography of the flowable composite. The structural integrity of the composite surfaces could be deteriorated by the applied bleach. Degradation of the resin matrix and the resin–filler bonds in the presence of bleach was reported by some authors [39,53] and could be supported by Steinberg et al. [52], who stated that resin composite surfaces are liable to chemical erosion in the presence of acidic and oxidizing bleaching agents. On the contrary, results of Dogan et al. [54] showed lower roughness for the bleached composite surfaces than the non-bleached, water-stored surfaces. In spite of the different types (micro-filled, micro-hybrid and ormocer) and brands of resin composites utilized in their study, they related their findings either to the finishing and polishing procedure that could result in some surface artifacts or to the resin erosion caused by the bleaching agents. However, among all the tested materials, the micro-filled composite with 40% of inorganic fillers was the most affected by the bleaching process. This finding could offer support to the results of the current study where composites with lower filler and higher resin matrix contents usually show more aggressive changes in their surface topography.

Results of the current study also indicated that topical fluoride application adversely affected the surface roughnesses of different types of resin composite. Surfaces of micro-hybrid, nano-filled, flowable and bulk-fill composites accordingly recorded higher Ras than non-treated and bleached surfaces of the same materials. Some researchers [52,55] referred this finding to the possible liability of resin composite surfaces to erosion in contact with acidic media like that existing in presence of APF preparation utilized in this study. In such an environment, the composite’s resin matrix usually decomposes and the filler particles could be removed from the material. Others [56,57] noticed a reverse relationship between the degree of surface degradation and the composite’s filler loading. Materials with lower filler loading usually show more degradation of their surface layers. This fact can explain why the fluoride-treated flowable composite of the current study (Table 1) showed the highest surface roughness among all the tested restorative materials.

The micro-hybrid type, at the same time, seemed more resistant to the erosive effect of the APF gel, as such surfaces showed lower roughness values than the other types of tested resin composites. Since the restorative composite degradation appears to depend on the filler size distribution through the number of matrix–filler interfaces, this finding could be referred to the material’s higher volumetric filler loading with different size distributions that allow minimal resin content to be exposed to the erosive acidic environment (Table 1) [5,6,7,8,9,10,11,12,13,14,15,16,17,18,19,20,21,22,23,24,25,26,27,28,29,30,31,32,33,34,35,36,37,38,39,40,41,42,43,44,45,46,47,48,49,50,51,52,53,54,55,56,57,58]. Han et al. [59] confirmed the previous explanation, as a direct relationship was observed between the distribution density of fillers on the surface resin and the resistance of resin composite surfaces to degradation.

The null hypothesis of this study suggested no adverse effects of both bleaching and fluoride application procedures on the surface hardness and surface roughness of the selected resin composite restorative materials. However, based on the study findings (Table 2 and Table 3), this null hypothesis should be rejected in part. The reason is that both bleaching and fluoride application procedures did show adverse effects on the surfaces of aged resin composite restorations; but some of the recorded alterations were too minimal to exceed the clinically acceptable values. Therefore, and in respect to the limitations of the current study, comparing the surfaces of freshly-prepared and aged composite resin restoratives is strongly recommended for further investigations to confirm the role of the increased composite polymerization on minimizing the adverse effects of both bleaching and fluoride preparations. Assessing of the wear resistance and the color changes of the aged resin composite restoratives following bleaching and fluoride applications is also recommended for future investigations.

## 5. Conclusions

Findings of the current study revealed that:Aged resin composite restorative materials can provide minimal surface alterations on successive bleaching and fluoride application; however, the flowable type of resin composites is the most affected by both clinical procedures.Although successive fluoride applications are deteriorating to the surfaces of the tested resin composites, repeated bleaching seems less lethal for the more viscous types.Following up of the existing restorations is advised following the bleaching and fluoride application procedures to determine the necessity of replacing the stained or worn restorations.

## Figures and Tables

**Figure 1 medicina-56-00476-f001:**
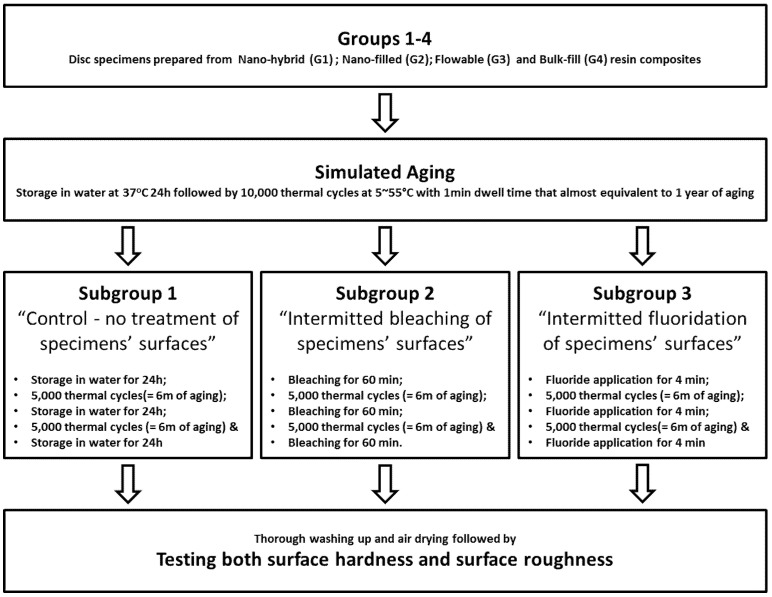
Schematic diagram showing different levels of the study, including the classification, surface treatment and testing of the constructed specimens.

**Table 1 medicina-56-00476-t001:** Materials used.

Product	Description	Composition	Manufacturer
Filtek Z250	Micro-hybrid universal composite restorative	**Matrix:** bisGMA, UDMA, TEGDMA, Bis-EMA	3M ESPE St. Paul, MN
		**Filler (78 wt%/60 vol%):** silica/zirconia. The filler particle size distribution is 0.01 µm to 3.5 µm with an average particle size of 0.6 µm.	
Filtek 350 XT	Nano-filled visible light-activated universal composite restorative	**Matrix:** bis-GMA, UDMA, TEGDMA, bis-EMA, PEGDMA resins.	3M ESPE St. Paul, MN
		**Fillers (72.5 wt%/55.6 vol%):** Combination of 20 nm non-agglomerated/non-aggregated silica filler; 4–11 nm non-agglomerated/non-aggregated zirconia filler, and 0.6–10 µm surface modified aggregated zirconia (4–11 nm)/silica (20 nm) clusters.	
Filtek 350 XT Flowable	Nano-filled visible light-activated flowable composite restorative	**Matrix:** bis-GMA, TEGDMA, procrylate resin	3M ESPE St. Paul, MN
		**Fillers (65 wt%/46 vol%):** Combination of 1–5 µm yetterbium triflioride fillers; 20 and 75 nm non-agglomerated, non-aggregated silica fillers and 0.6–10 µm surface modified aggregated zirconia (4–11 nm)/silica (20 nm) clusters.	
Filtek Bulk-fill	Nano-filled visible light-activated posterior composite restorative.	**Matrix:** AUDMA, UDMA and 1, 12-dodecane-DMA.	3M ESPE St. Paul, MN
		**Filler (76.5 wt%/58.4 vol%):** combination of a non-agglomerated/non-aggregated 20 nm silica filler, a non-agglomerated/non-aggregated 4 to 11 nm zirconia filler, aggregated zirconia/silica cluster filler (comprised of 20 nm silica and 4 to 11 nm zirconia particles) and a ytterbium trifluoride filler consisting of agglomerate 100 nm particles.	
Opalescence Boost PF	Chemically-activated neutral (pH 7) in-office bleaching agent	**Barrel 1:** 1.1% sodium fluoride and 3% potassium nitrate, along with a unique chemical activator.	Ultradent Products Inc. South Jordan, UT
		**Barrel 2:** Hydrogen peroxide.	
		After mixing the final hydrogen peroxide concentration is 40%.	
Gelato APF gel	Acidulated phosphate fluoride gel	**Active ingredients:** 2.09% Sodium fluoride and Hydrofluoric acid providing 1.23% fluoride ions.	Deepak Inc. Miami, FL
		**Inactive ingredients:** Flavor, phosphoric acid, sodium saccharin, xylitol, citric acid, sodium benzoate, water, titanium dioxide polysorbate 20, xanthan gum, magnesium aluminum silicate, FD&C red # 40.	

**Table 2 medicina-56-00476-t002:** Surface hardness (VHN) values of resin composites in different subgroups.

Surface Treatment	Resin Composites			
	Conventional			Bulk-Fill (G4)
	Micro-Hybrid (G1)	Nano-Filled (G2)	Flowable (G3)	
No-Treatment (SG1)	68.85 ± 3.54 ^A,1^	55.26 ± 4.05 ^B,1^	49.83 ± 9.48 ^B,1^	48.77 ± 5.29 ^B,1^
Bleaching (SG2)	68.97 ± 4.59 ^A,1^	57.05 ± 7.12 ^B,1^	41.11 ± 8.19 ^C,1,2^	41.88 ± 7.76 ^C,1^
Fluoride (SG3)	53.90 ± 5.69 ^A,2^	51.31 ± 4.56 ^A,1^	38.66 ± 3.64 ^B,2^	39.96 ± 2.55 ^B,1^

Different superscript numbers in each column (group) indicate significant differences between subgroups (Tukey’s, *p* < 0.05). Different superscript letters in each row (subgroup) indicate significant differences between groups (Tukey’s, *p* < 0.05).

**Table 3 medicina-56-00476-t003:** Surface roughness (nm) values of resin composites in different subgroups.

Surface Treatment	Resin Composites			
	Conventional			Bulk-Fill (G4)
	Micro-Hybrid (G1)	Nano-Filled (G2)	Flowable (G3)	
Non-Treatment (SG1)	17.34 ± 2.90 ^A,1^	14.64 ± 3.15 ^A,1^	8.32 ± 1.26 ^A,1^	10.18 ± 2.69 ^A,1^
Bleaching (SG2)	18.68 ± 1.47 ^A,1^	19.81 ± 2.18 ^A,1^	45.36 ± 6.86 ^B,2^	21.50 ± 2.68 ^A,1^
Fluoride (SG3)	39.90 ± 4.36 ^A,2^	58.20 ± 10.55 ^B,2^	76.94 ± 4.97 ^C,3^	63.24 ± 7.44 ^B,2^

Different superscript numbers in each column (group) indicate significant differences between subgroups (Tukey’s, *p* < 0.05). Different superscript letters in each row (subgroup) indicate significant differences between groups (Tukey’s, *p* < 0.05).

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
