# Peer review of "Influences of Successive Exposure to Bleaching and Fluoride Preparations on the Surface Hardness and Roughness of the Aged Resin Composite Restoratives"

_medicina, 2020, doi:10.3390/medicina56090476_

Round 1
Reviewer 1 Report
Dear Authors,
Now the manuscript could be accepted for publication, all my comments have been addressed. I suggest to read a cite a recent inherent manuscript about fluoride: https://www.mdpi.com/2313-7673/5/3/41
Thank You
Kind Regards
Author Response
Thanks. I do appreciate your valuable comments. The reference you've suggested has been considered and cited within the article (#Ref # 35).
Thanks once more for the great effort you did in revising this manuscript.

Reviewer 2 Report
The article entitled "Influence of Successive Exposure to Fluoride and Bleaching Agents on Surface Hardness and Roughness of Contemporary Resin Composite Restoratives" presented a clear and well-written form. New and actual references have been entered, and the material methods section has been corrected. Therefore, the article is accepted in the present form.
Author Response
Thanks a lot, I do appreciate the effort you did in revising this manuscript.
Reviewer 3 Report
The revised version has addressed my comments and suggestions for improvement. It is now ready to be published.
Author Response

(The authors gave the same response as above.)
